# Fungi That Promote Plant Growth in the Rhizosphere Boost Crop Growth

**DOI:** 10.3390/jof9020239

**Published:** 2023-02-10

**Authors:** Afeez Adesina Adedayo, Olubukola Oluranti Babalola

**Affiliations:** Food Security and Safety Focus Area, Faculty of Natural and Agricultural Sciences, North-West University, Private Bag X2046, Mmabatho 2735, South Africa

**Keywords:** abiotic factors, biotic factors, crop protection, crop science, phytohormones, plant growth, IR, soil microbial ecology, sustainable agriculture

## Abstract

The fungi species dwelling in the rhizosphere of crop plants, revealing functions that endeavor sustainability of the plants, are commonly referred to as ‘plant-growth-promoting fungi’ (PGPF). They are biotic inducers that provide benefits and carry out important functions in agricultural sustainability. The problem encountered in the agricultural system nowadays is how to meet population demand based on crop yield and protection without putting the environment and human and animal health at risk based on crop production. PGPF including *Trichoderma* spp., *Gliocladium virens*, *Penicillium digitatum*, *Aspergillus flavus*, *Actinomucor elegans*, *Podospora bulbillosa*, Arbuscular mycorrhizal fungi, etc., have proven their ecofriendly nature to ameliorate the production of crops by improving the growth of the shoots and roots of crop plants, the germination of seeds, the production of chlorophyll for photosynthesis, and the abundant production of crops. PGPF’s potential mode of action is as follows: the mineralization of the major and minor elements required to support plants’ growth and productivity. In addition, PGPF produce phytohormones, induced resistance, and defense-related enzymes to inhibit or eradicate the invasion of pathogenic microbes, in other words, to help the plants while encountering stress. This review portrays the potential of PGPF as an effective bioagent to facilitate and promote crop production, plant growth, resistance to disease invasion, and various abiotic stresses.

## 1. Introduction

The population of humans globally is anticipated to rise to approximately 9.7 billion from the issues encountered nowadays, such as high birth rates, etc., by the year 2050, as estimated by the Food and Agricultural Organization of the United Nations (FAO) [1,2,3]. The population growth, expected to rise by 34% in several years, will affect food prices [4]. From another point of view, the production of food crops cannot be expected to increase at this rate. Therefore, there is a need to improve agricultural production to equilibrate the socioeconomic challenges that might erupt due to population growth. The production of foods and other agricultural products should increase from the current food production by 70% to sustain the population globally [5]. To achieve this goal, plant disease must be inhibited or eradicated to promote food security.

Crop plants such as tomatoes, cucumbers, and carrots are primary vegetables consumed by humans and animals globally because of the essential vitamins, carotene, and minerals they contain [6]. However, these crops, including tomatoes, cucumbers, carrots, oranges, etc., are affected by phytopathogens, such as powdery mildew caused by *Oidium neolycopersici* on tomatoes [7], downy mildew caused by *Pseudoperonospora cubensis* on cucumbers [8], powdery mildew *Erisyphe heraclei* on carrots [9], and soft rot on oranges caused by *Penicillium digitatum*, and *Fusarium oxysporum* [10], thereby reducing their products globally. Phytopathogens are microorganisms that produce phytopathogenic toxins that destroy crop plants, thereby reducing the production of crops for human consumption [7]. In addition, they cause various diseases, and the microbial agents responsible include soft rot (*Fusarium oxysporum*, *Aspergillus niger*) [11,12], early blight (*Alternaria solani*) [13], and powdery mildew (*Oidium neolycopersici*) [14]. Aside from the effect of phytopathogens, certain chemical derivatives meant to inhibit the existence of pests and diseases or to boost the fertility of agricultural soil are potentially detrimental to the health of humans and animals once fed the crop to which the chemical has been applied [10], thereby leading to less production. Due to these problems encountered, researchers have worked out alternative means that can potentially inhibit the growth of phytopathogens and replace agrochemicals once applied on farmland.

Plant-growth-promoting fungi (PGPF) resident microorganisms residing in the rhizosphere soil of various crop plants, in which tomato plants are not left out, are one of the effective and ecologically friendly derivatives scientists investigated to inhibit or eradicate plant diseases. They were known for improving the defense mechanisms and growth in plants [15]. PGPF applications suppress the usage of agrochemicals and likewise prevent plants from biotic and abiotic stresses [16]. Various studies reported the following PGPF genera (*Gliocladium*, *Penicillium*, *Phoma*, *Phytophthora*, *Rhizoctonia*, *Talaromyces*, *Trichoderma*) to improve tomato [15], orange [10], apple [17], pear [18], cucumber [19], carrot [20], and other plants’ growth and further promote the plants’ innate immunity and the production of various necessary secondary metabolites by the plants. PGPF perform the following functions in plants: antagonistic or biocontrol potential by competing for space and nutrients [10], growth hormone production [21], mineral solubilization, mycoparasitic and saprophytic resistance, root colonization, and induced systemic resistance (ISR) in plants [22]. Aside from the above roles mentioned, PGPF suppress the invasion of phytopathogens on tomato plants as well as other crop plants, they contribute to the improvement of nutrients in the soil, and they produce 1-aminocyclopropane-1-carboxylate (ACC) deaminase and other phytohormones to reduce the production of ethylene in the plants [23].

The low-yield production of tomatoes and other crops is actualized by biotic and abiotic stresses that are of significant concern that reduce soil fertility [24]. Drought, high and low temperatures, high winds, and soil salinity are prominent abiotic stresses affecting the production of crop plants. Destructive environmental stresses that implement reductions in farmland and agricultural produce are soil salinity and soil solidification.

The biotechnological technique is sustainable and effective in stimulating the interaction between PGPF and plant root exudates, which is required to improve the abundant production of crops and as well to promote soil health [25]. The PGPF found in the rhizosphere can improve the level at which plants respond to environmental stresses and promote crop production. In this study, crop plants are observed as superorganisms that count so much on PGPF for purposeful roles and characters. However, various studies have reported the association between PGPF and crop plants to improve the growth, health status, and production of healthy crops [7,21,26]. Therefore, crop plants can exploit PGPF in line with their advantages by selectively stimulating PGPF with the potential to induce the growth and development of tomato plants.

## 2. Microbial Communities’ Beneficial Association in the Rhizosphere Soil of Crop Plants

PGPF can change the bacteria’s physiological features in the soil [21]. The *Pseudomonas* spp. population in the rhizosphere depends on the roles it carries out and the activity of the ectomycorrhizosphere [27]. The bacteria can survive on the fungi hyphae, most notably when they can produce a biofilm or secrete type III secretion systems (ITSSs) as the mechanism of survival on the hyphae aside movement. The exudates produced by the roots of mycorrhiza are a substrate for the conventional growth of bacteria, and the bacteria activate changes and the availability of both *Arbuscular mycorrhizal* fungi and tomato plants to nutrients [28]. The interactive association between AMF and *Bradyrhizobium japanicum* has been revealed to help stimulate proper growth in most tomato plants [29]. AMF and PGPF have also been reported to improve the growth of cucumbers and likewise inhibit damping off [30]. PGPF (*T. viride* and *P. chrysogenum*) have been reported for their efficacy on oranges to prevent post-harvest diseases [10].

Endosymbiosis is an association between bacteria and class Basidiomycetes, despite little interaction occurring in class Ascomycetes. Intracellular association has challenging advantages for plants due to the different number of growth-promoting enzymes affected during these associations. The symbiotic association occurring between fungi and bacteria is not only restricted to the benefits obtained in tomato plants, but also the potential to protect bacteria from the activities of antibiotics, as in *Piriformospora indica* [31]. With bacterial presence, the ability of fungi to involve in endosymbiosis assists some of the phyla in completing their life cycle [32]. Thomas and Singh [33] reported the endosymbiosis association that was characterized between Basidiomycetes and Zygomycetes, which further revealed how numbers of bacteria could be sheltered by Glomeromycota fungi spores, thereby retaining their existence in their habitat.

### 2.1. PGPF Improve Crop Production under Biotic Stress

The soil microbiome has a beneficial association with plants, thereby improving the plants’ well-being and productivity. These microbiomes are found in the rhizosphere of crop plants, among which are algae, archaea, arthropods, bacteria, chlamydia, fungi, nematodes, viruses, oomycetes, protozoa, etc. [7]. Various fungal classes, including Ascomycetes, Basidiomycetes, and Deuteromycetes, can produce the PGPF trait (PGPT) of plant growth (Figure 1). For the treatment of tomato seeds, PGPF have been used to improve the growth and production of tomatoes, likewise stimulating the elongation of shoots and roots [7]. The crop plant’s potential to photosynthesize and improve crops can be promoted by inoculating PGPF [34]. PGPF can likewise induce the stimulation of secondary metabolites from the crop plant. The secondary metabolite compounds that protect tomato plants from phytopathogens can be attained in various ways. PGPF stimulate antimicrobial agents and can be a biocontrol agent to control or inhibit the growth of phytopathogens. These microorganisms also stimulate systemic resistance (SR) to safeguard plants. Induced systemic resistance (ISR) is the process by which plants antagonize the activities of phytopathogens where PGPF are active by inducing plant defenses [35].

The process by which PGPF promote the production of crops includes inhibiting the invasion of phytopathogens such as bacteria, fungi, nematodes, and oomycetes. Disease occurrences as a result of these phytopathogen attacks cause a loss of crop production globally. Tolerating biotic stresses is essential for completing the growth phase. To eradicate the problems encountered by biotic stresses, crop plants have formulated many methods for adapting to the environment for living [36]. Crop plants respond to external stress, induced, and then stimulated cellular responses. This runs to differential transcriptional changes, engaging the plant to tolerate various stresses. The signaling tracts perform an essential function and play a connecting link between environmental stress and creating a biochemical and physiological response.

Plants are mostly subjected to various activities of phytopathogens. For plants to keep themselves safe, PGPF inhabiting the rhizosphere soil and those embedded in plant tissues are a defense mechanism to safeguard themselves from activities induced by phytopathogens’ invasion [37]. The incursion of the plant cell wall subjects the plant spoilage organisms to the cell membrane of the plants where they come across extracellular surface receptors that identify pathogen-associated molecular patterns (PAMP) [38].

The identification of phytopathogens at the cell surface activates PAMP-triggered immunity (PTI), which commonly inhibits infection before the phytopathogen invades the entire plant [38,39,40].

#### 2.1.1. Phytohormone Production

Phytohormones perform an important function in controlling the growth of crop plants and their defense mechanisms [23]. IAA is a prominent auxin hormone that promotes the growth of crop plants’ shoots and roots. *Penicillium janczewskiy* was reported to stimulate inhibition against *Rhizoctonia solani*, a phytopathogen causing stem rot, by producing IAA [41]. Auxin precursor’s engagement in these processes has not been confirmed, but the growth of plant roots is improved systematically with the ascertained effects of the stimulation of fungal-based auxin. PGPF, including *Aspergillus*, *Mortierella*, *Talaromyces*, *Fusarium*, *Penicillium*, and *Trichoderma*, reported being isolated from the root-soil to produce IAA from plants and contribute to the development of tomato plants [19]. Some phytohormones control the development of seeds and plant roots and shoots alongside IAA and gibberellin (GA). Likewise, the application of *Cladosporium* sp. to tomato plants improved their growth due to the production of GA [42], and it has been proposed that the production of GA performs an important function in the colonization of pea plants. According to Meena, et al. [35], other PGPF evoked cytokinin production, which supported growth in plants. Likewise, Attia, et al. [15] reported how PGPF stimulate abscisic acid and promote tomato plants’ growth under biotic and abiotic stresses.

#### 2.1.2. Plant Growth Promotion

Various fungi inhabiting the rhizosphere of tomatoes and other crop plants contribute positively to the growth of the plants. The following fungi, including *Aspergillus*, *Penicillium*, *Rhizoctonia*, *Talaromyces*, *Trichoderma*, etc., are involved in plant growth promotion, as reported in Table 1. In contrast to the above fact, there are some species of fungi, including *Aspergillus* (*A. wentii*, *Aspergillus versicolor*, and *A. niger*) [10,43], *Fusarium* spp. (*F. oxysporum*) [44], and *Penicillium* spp. (*P. citreonigrum*, *P. citrinum*, *P. digitatum*, and *P. janthinellim*) [45], that cause crops diseases. Various studies have reported how *Trichoderma* species (*T. viride*, *T. harzianum*) and *Penicillium chrysogenum* promote the growth of tomatoes and other crop plants [46]. The study further revealed how these PGPF are unique in the host plants associated with growth promotion. It has been reported that various PGPF residing in the rhizosphere, including sterile black fungi (SBF), sterile dark fungi (SDF), and sterile red fungi (SRF) in crop plants such as barley, bromegrass, chickpeas, clovers, corn, lupine, peas, and wheat, contribute to and enhance the growth promotion of these crop plants [41]. The above report coincides with the study that revealed how *Trichoderma* spp. and *Penicillium* spp. improve the development of rice crop plants in no more than a month [47], although other studies claimed that PGPF could take a long time before the growth of the crop plants can manifest.

Several studies have reported the importance of PGPF in greenhouse and field experiments as they promotes the growth of crop plants employed in the studies [53]. The conidial suspension of *A. niger*, *P. citrinum*, and *T. harzianum* for the treatment of chickpea seeds has substantially enhanced the plant’s growth [41]. Moreover, Brassica, chili, and cucumber plants’ growths were reported to be enhanced by the following PGPF: *Chaetomium* spp., *Exophiala* sp., and *Talaromyces* spp. [50]. *Penicillium* sp. enhanced chili, sesame, and tomato plants’ growths [57]. However, *T. koningi* enhanced the growth as well as the stimulation of phytoalexin, and this species of PGPF was able to establish itself in the colonization of *Lotus japonicus* plant roots. Aside from the facts that have been revealed above, the potential of green plants to synthesize sunlight (photosynthesis) is increased when inoculated with PGPF, as observed with some plants. The results show how the chlorophyll content increases, thereby improving the plant’s potential to photosynthesize, further enhancing plant growth [58]. The colonization of PGPF in plant roots enhances plant root growth and promotes plant growth. In comparison to the above study, some PGPF have been revealed not to have the potential to colonize the roots of plants to promote development and crop production [59]. In the case that colonization is achieved, the proficiency of the PGPF will be observed at the upper, mid, or lower region of the plant roots. Therefore, the process which involves the colonization of roots by PGPF may not be a measure for the improvement of plant growth, which is regarded as a non-homogenous process.

PGPF treatment for the blooming of flowers performs an important function in the development of fruit in agricultural and ornamental plants [60]. When tomato and Arabidopsis plants were treated with *T. harzianum* and *P. chrysogenum*, there came a sudden effect on the bloom size and numbers of the flowers. Other fungi species such as *Phoma* sp. treatment enhanced the height of cucumber plants and the number of leaves and fruits produced in greenhouse experiments [61]. *Rhizoctonia* sp., *R. delemar*, *T. longibrachiatum*, and non-sporulating fungi improved the produce obtained from different crop plants [41].

From PGPF, the crude extracts and metabolites extracted have been used to evoke plant growth and to induce resistance against phytopathogen invasion [62]. These extracts may be oligosaccharides, proteins, and sphingolipids that positively impact plant growth characteristics. From PGPF, the proteins extracted enhanced the seed quality features prevalently in pearl millet and musk melon plants, unlike the growth of the control plants [63]. Likewise, compounds released by PGPF carry out an important function in plant growth and development. The application of terpenoids, a volatile substance obtained from *T. wortmannii* and *Phoma* sp. evokes the growth of tomato plants. The study reveals how the compound released by PGPF enhances the growth of lateral roots, stimulates flowering activities, and increases the chlorophyll concentration in the leaves [19].

#### 2.1.3. Nutrient Mineralization

PGPF stimulate enzymes, such as inorganic acids, organic acids, phosphatases, and phytase, for complex organic phosphorous solubilization and mineralization into various small molecules [16]. The mineralization of nutrients in the soil mediated by PGPF carries out a special function in plant growth improvement. It reduces the breakdown of the substrate into a soluble molecule that can be assimilated by plants. Phosphorus (P) is an important macroelement that promotes plant growth. Fungus (*Ceriporia lacerata*) was reported by Sui, et al. [64] to solubilize the phosphates by the activity of organic acid and phytase. In addition to this, *Aspergillus awamori* and *Penicillium digitatum* were also reported to solubilize phosphate [65]. Moreover, *T. viride* demonstrated how to promote the content of P, N, K, and organic carbon in the soil [66]. Among the PGPF, *Trichoderma* sp. was reported to promote the absorption of minerals and nutrients, mainly, Fe, K, N, and P [67].

Cyanide is an important secondary metabolite acquired during the early fixed period of plant development. Various amino acids, including glutamate, cysteine, glycine, and methionine, are the forerunners in the oxidative decarboxylation processes [68]. *A. niger*, isolated from tomato rhizosphere soils, produces HCN and ammonia that adequately enhance the growth of tomato plants [69]. *Trichoderma* spp. are known for their potential to hydrolyze ACC into ammonia for plant growth [70].

#### 2.1.4. Synergistic Potential of Fungi as Biological Control Organisms

Biocontrol, also known as antagonism, is a process by which rhizospheric microorganisms (fungi) inhibit or eradicate the growth of spoilage organisms in plants. The PGPF biocontrol potential is attributed to the competition for nutrients that the organisms feed on, the production of antibiotics against pathogens, and microbial predation. The process of antagonism is also attributed to the stimulation of lytic enzymes such as chitinase protease and glucanase.

The predation potential of PGPF is the initiation of the chitinase gene that reduces the growth of phytopathogens. *Gliocladium virens* is an example of a PGPF that produces Gliovirin, an antibiotic that inhibits the growth of *Pythium ultimum* [71]. From the rhizospheric soil of a plant, Murali, et al. [41] reported how nonpathogenic strains of *F. oxysporum* were isolated, thereby restricting the invasion of spoilage organism strains of the same organism, *Fusarium*, through competition for nutrients, the inhibition of chlamydospore growth, and contention for space, especially in the roots, where the infection persisted. Moreover, *T. harzianum* inhibited the growth of the following species of fungi: *F. oxysporum*, *Pythium* sp., *R. solani*, and other phytopathogens, as reported by [72]. Fantastically, various fungi isolated from the rhizosphere of healthy crop plants [73] and trees such as the Moringa tree (*Moringa oleifera*) [10] have displayed antagonistic activities against the growth of phytopathogens. Biocontrol activity is not a peculiar measure for the initiation of resistance. As a result, other PGPF found in the rhizosphere soil do not possess antagonistic properties that induce resistance in plants against phytopathogenic diseases [21,22]. As a result, there are other features, unlike antagonism, that can carry out various roles in the resistance against phytopathogen invasion.

#### 2.1.5. Induced Resistance

Induced resistance (IR) is a distinguished process of promoting the immunity of plants. It is derived after treatment with an elicitor or inducer that acts against biotic stresses constrained by microbes (Figure 2) that include bacteria, fungi, nematodes, parasites, etc. IR is classified based on the inducer and the potential to manipulate the route of defense signaling, and there are two types: induced systemic resistance (ISR) and systemic acquired resistance (SAR). ISR is a process that involves inhibiting the disease invasion in plants invented by certain phytopathogens. The initiation of ISR in plants upon phytopathogen attack is consistently revealed in the plant organs that are distanced from the inducing area [62]. On the other hand, the ability of PGPF in the priming of the plant roots does induce resistance in the shoots—the upper part of the plant [41]. This systematical induction of resistance upon PGPF treatment is regarded as a safe and effective way to promote the growth of crop plants and the cultivation of abundant yield.

The treatment of plants with PGPF, such as non-sporulating fungi, *Trichoderma* spp., and *Penicillium* spp., elicits ISR by inhibiting the development of soil-borne phytopathogens in crop plants [41]. Puri, et al. [74] reported how *P. chrysogenum* treatment induced systemic resistance to *Verticulum* wilt in cotton caused by *Verticillium dahlia.* Elsharkawy, et al. [75] revealed in their study how *P. simplicissimum* induced ISR in cucumber plants. *Trichoderma* sp. has shown its potential to induce ISR on phytopathogen invasion in crop plants [76]. In tomatoes, *T. harzianum* and *T. viride* induced systemic resistance against *Botrytis cinerea*, causing grey mold disease [77], and *Acremonium sclerotigenum* treatment was also reported by Llorens, et al. [78] to induce systemic resistance against *P. syringae* pv. in tomatoes.

Systemic acquired resistance (SAR) is a form of induced resistance that expresses a discrete signal transduction pathway. This pathway is made up of the stimulation of pathogenesis-related proteins and carries out a particular function in promoting plant defense against phytopathogens [62]. In tomatoes, the stimulation of SAR substantially inhibits the prevalence of infections caused by spoilage organisms, including the PGPF *Metarhizium brunneum* and *Beauveria bassiana* [79]. Under low nutrition, *Trichoderma asperelloides* promotes the effect of acquired resistance in Arabidopsis [41]. Therefore, PGPF-mediated SAR activates an ISR in plants via protein activation against phytopathogens.

Various metabolites obtained from PGPF effectively contribute to resistance in crop plants. This aspect of elicitor-inducible proteins corresponds to contributing to resistance to various diseases [80]. Certain elicitors such as glycoprotein have been extracted from *P. cinnamomic* [81], *Cladosporium fulvum* [82], *C. lindemuthianum*, and *R. solani* [83], which displayed the activity of inducing resistance in crop plants. In tobacco plants, an elicitin protein extracted from *Phytophthora* sp. to stimulate defense mechanisms [62] and also from *T. virens*, a proteinaceous non-enzymatic elicitor, was obtained that effectively activated defense responses in plants and likewise stimulated induced systemic resistance against foliar phytopathogens [41]. Strong elicitors such as oligosaccharides were extracted from PGPF that improve the activation of plant defenses against phytopathogens. Soft rot disease-causing organisms on tomatoes and other crops, *F. oxysporum*, produce cerebrosides, mainly glycosphingolipids, which have revealed resistance against wilt and soft rot diseases in tomato plants. This cerebroside treatment in greenhouse experiments also inhibited the proliferation of anthracnose disease in chili plants [84].

#### 2.1.6. Defense Mechanisms

The important mechanism plant spoilage organisms pass through to enter the host cell is the plant cell wall. PGPF, however, stand to enrich the plant cell defense to avoid the invasion of pathogens via improving the deposition of defensive cell wall components, mainly cellulose, lignin, and phenol. One of the most fundamental problems plants encounter is the invasion of phytopathogens. Plants stimulate different forms of mechanisms of defense to resist disease invasion. The resistance process signifies the stimulation of an allergic reaction (AR) and the biological synthesis of antimicrobials such as phytoalexins that contribute to plant reduction. The AR results in the mortality of plant cells mainly occurring at the point of infection and happens to reduce the proliferation of phytopathogens. The phytopathogen is resisted during the interaction with the host plant, and brown necrotic spots are observed at the point of the infection. The basic activity of AR is the immense stimulation of ROS compounds, amounting to the stimulation of ROS, such as hydrogen peroxide (H_2_O_2_), superoxide, and hydroxyl radicals. In plant cell membranes, the accumulation of H_2_O_2_ is a result of the stimulation of ROS at the point of infection, therefore reducing the proliferation of the phytopathogens [85].

The produced ROS is allergic when produced at high concentrations, causing a fatal effect on beneficial microbes and plant cells. Yet, this effect is produced by antioxidant enzymes in that its performance is increased after PGPF introduction following biotic and abiotic stress [56]. Various studies have reported the accumulation of callose during phytopathogen infection, activating resistance during the interaction of plants and phytopathogens. The interaction also occurs at the cost of pretreatment processes with various chemical inducers that activate natural immunity in plants. Kamble, et al. [86] explained how *T. harzianum* promoted callose accumulation in cucumber seedlings. Likewise, the accumulation of callose in *A. thaliana* roots was enhanced by *T. harzianum* [87]. The treatment of pearl millet seedlings with *T. hamatum* revealed a high concentration of callose and lignification accumulation during the infection period with a downy mildew phytopathogen [86]. The *T. atroviride* strain was reported to improve the callose accumulation in cucumber plants after the invasion of *Rhizoctonia solani* [41]. Furthermore, the treatment of the invasion of *Talaromyces funiculosus* promoted the accumulation of callose on the external layer of *C. capsici* seedlings. In the phenylalanine or tyrosine metabolic pathway, lignin is produced in plant cells, which carries out the potential activity as a barrier against phytopathogens [88]. It activates defense mechanisms in plants after the phytopathogen has invaded through the accumulation of lignin on the wall, especially at the point of phytopathogen invasion.

##### Biochemical Defense

The congenital immunity of plants against phytopathogen invasion is connected to biochemical defense mechanisms, unlike structural and histological changes, as biochemical modifications take place during the procedure. The associations that occur between phytopathogens and the plant host promote modifications in the metabolism of cells and the potentials of enzymes, such as β-1,3 glucanase, lipoxygenase (LOX), peroxidase (POX), phenylalanine ammonia-lyase (PAL), polyphenol oxidase (PPO), and superoxide dismutase (SOD) [39]. The treatment of seeds with PGPF, such as *Aspergillus* spp., *Fusarium* sp., *Rhizoctonia* spp., *Trichoderma* spp., *Penicillium* spp., *Phoma* spp., *Pythium* spp., *Talaromyces* spp., and non-sporulating fungus, promoted the activity of POX prevalence in fruits such as cucumbers, chilis millet, and pearls [41]. Introduction of *Trichoderma* spp. promoted the activities of chitinase and peroxidase in cucumber plants’ leaves and roots [89]. The high rate of PPO was a result of beneficial PGPF in the banana plant when *F. oxysporum* was introduced into the plant’s roots [41]. The PPO process in cucumber plants increased in level after being treated with *Phoma* spp. and a non-sporulating fungus [90]. It has also been observed in the *Trichoderma* strain, as it increased the PPO enzyme activity in chilis, moong beans, and pigeon peas after being inoculated with *Alternaria alternate*, *C. capsici*, and *F. oxysporum* [91].

The activity of LOX in pearl millet increased using *Trichoderma* sp. [77], and the colonization of millet plant roots with these isolates increased the rate of enzymes such as chitinases, peroxidases, and β-1-3-glucanases [41]. IR revealed how primed expression is accompanied by defense-related genes after being inoculated with *Fusarium oxysporum* [92]. Upon introduction of *Colletotrichum orbiculare* with PGPF such as Phoma spp. and nonsporulating fungus, the chitinase activity improved in cucumbers [41]. The introduction of metabolites obtained from fungi induced defense responses in plants by applying defense enzymes, such as LOX, PAL, POX, and PPO [93].

##### Defense Signaling

Various defense enzymes, such as signal molecules and PR proteins, are produced in plants to carry out antimicrobial processes during local and systemic defense responses. When PGPF-mediated induced resistance was investigated, the specific result produced by defense enzymes piled up. Defense responses activated by pathogenic and beneficial microbes control the defense response processes, despite the phytohormones ethylene (ET), jasmonate (JA), and salicylic acid (SA) carrying out specific functions [94]. The specific indication reveals the cross-communication between the phytohormone tracts to proportion the plant defense responses situated on the phytopathogen invasion. JA or ET systemic resistance was triggered in cucumber plants when treated with *T. asperellum* and associated with the PR gene expression to act against phytopathogen invasion [95].

A few studies have revealed the potential of the SA-dependent response after being treated with *Trichoderma* sp. [96,97]. The treatment of tomato plants with *Trichoderma* sp. revealed how the JA-responsive gene expression increases and activates systemic resistance to inhibit the phytopathogen *B. cinerea* [96]. Yu, et al. [98] showed how *Trichoderma* sp., triggering ISR, partakes in ET and JA signaling. Furthermore, the stimulation of both SA and JA signaling has been reported for a few *Trichoderma* strains. Some species of *Trichoderma* were reported to decompose cellulose in plants following enzyme activities [99], which stimulates ISR activities in plants, such as beans, corn, cucumbers, limes, tobacco, etc., by inducing ET or JA defense signaling.

### 2.2. PGPF Improve Crop Production under Abiotic Stress

Pests and diseases are known for their role in reducing cultivation globally, but probably not as significant as the role of abiotic stress. Crop plants undergo abiotic stress when the conditions of the environment become extremely strong (Figure 3) compared to the optimum level for growth and reproduction [100]. Biological stress is defined as an environmental factor stimulating the strain in living organisms that have been injured. These biological strains are affected by chemical or physical changes as a result of various stresses to which the microorganisms have been exposed [101]. Most factors contributing to stress in crop plants originate from climatic conditions, such as extreme temperature, extreme or no rainfall, radiation, and salinity (mostly common to desert regions where ions build up in the soil).

Crop plants undergo problems posed by abiotic stresses. These stresses include high drought, floods, salinity, temperature, and heavy metal aggregation that impact crop plants’ growth and production. Farmlands undergo some of the abiotic stresses as listed above, reducing crop production to 70%. The changes observed in the climate jeopardize the future loss of the production of crops, specifically cereal crop plants, thereby emanating challenges in the food security sector. In the years 1990 to 2013, several farmlands revealed a high saline content (up to 37%), as reported by Waaswa, et al. [102]. PGPF have revealed their alternative option to improve the growth promotion of plants under these abiotic stresses (Figure 1) as a result of root production, solubilization of minerals from fossils, and the production of secondary metabolites. PGPF are embedded in the rhizosphere of crop plants. They can improve the immunity of plants and further support plant growth under stress (Table 2). In tomato plants, *T. harzianum* promoted the early germination of seeds and strength with the growth of shoots, weight, and root length under abiotic, biotic, and physiological conditions [41]. The growth of cucumber plants is supported by *Trichoderma asperellum*, which induces siderophore production under salinity conditions. *Penicillium* spp. has also been reported to tolerate saline conditions in sesame plants [103]. *Trichoderma* spp. induced the toleration of abiotic stress potential against biotic pressure inflicted by phytopathogens [39]. The possibility of drought tolerance in maize plants has been improved by the introduction of *T. atroviride* with the increased antioxidant enzyme. The introduction of *T. harzianum* reduces the salinity stress in Indian mustard plants with the increased antioxidant enzyme. In *Theobroma cacao*, there was the activation of plant growth and improved drought conditions upon treating the plant with *T. hamatum*.

The accumulation of heavy metals in Arabidopsis plants showed how PGPF, including *Aspergillus*, *Alternaria*, *Microsphaeropsis*, *Mucor*, *Peyronellaea*, *Phoma*, and *Steganosporium*, mitigated heavy metals in the roots of the plants. Ram, et al. [104] revealed how *T. harzianum* promotes salinity, the heat shock protein, osmotic pressure, and oxidative stress status in *A. thaliana* plants. In *N. tabacum* plants, *T. virens* increased the cadmium tolerance potential with a decreased lipid peroxidation and the promotion of the antioxidant enzyme mechanism [105]. In addition, the drought tolerance potential in *N. tabacum* plants increased due to *T. harzianum* introduction with an increment in water content with a reduction in the transpiration level [106]. Moreover, it increased antibiotic production, mycoparasitism, and antagonism and stimulated ISR.

Plants’ cell walls are fortified by PGPF, thereby preventing nutrient loss during abiotic stress [24]. The accumulation of callose escalated during stress conditions, thereby increasing the plugging of the sieve pores, cell membrane accumulation, and cell wall modification. Lignin is a complex polymer that partakes in plant defense against phytopathogens and pests, and its actions are improved in plants when stimulating the tolerance to various abiotic stresses, such as drought, heavy metals, extreme temperature, and salinity [107]. PGPF have the potential to tolerate salinity conditions due to an increase in sterol content for the alteration of fatty acid enzymes. So, they are regarded as salt-tolerant and loving microbes due to the underlying osmotic condition.

**Table 2 jof-09-00239-t002:** Plant-growth-promoting fungi’s potential against abiotic stresses in crop plants.

PGPF	Crop Plant	Action against Abiotic Stresses	References
*Fusarium*, *Gliocladium*, *Penicillium*, *Phytophthora*, *Phoma* spp., *Rhizoctonia*, *Talaromyces*, and *Trichoderma*,	*Oryza sativa* and *Zea mays*	Stimulate phytohormones, defensive compounds, and defense-related enzymes that inhibit phytopathogen invasion, thereby helping the plants against biotic and abiotic stresses	[41]
*Epichloë typhina* and *Curvularia protuberate*	*Solanum lycopersicum*	Improve the survival of plants by reducing the negative impacts of biotic and abiotic stresses, such as drought, salinity, extreme temperatures, heavy metal toxicity, and oxidative stress	[108]
*Cunninghamella bertholletiae*	*Solanum lycopersicum*	Inhibit the occurrence of salinity, drought, and heavy metal stresses	[109]
*Arbuscular mycorrhizae*	*Triticum* spp.	Improve stress tolerance, thereby contributing to the plant growth	[24]
*Arbuscular mycorrhizal*	*Pinus edulis*	Carry out a particular function in reducing abiotic stresses, therefore enhancing plant growth	[110]
*Trichoderma virens*	*Ceratocystis paradoxa*	Biologically control pathogens, such as *Fusarium verticillioides*, *Colletotrichum falcatum*, *Ceratocystis paradoxa*, and *Xanthomonas albilineans*, in tolerance of abiotic stresses	[111]
*Arbuscular mycorrhiza* fungi	*Melissa officinalis*	Drought stress was controlled by the fungi after being inoculated by promoting photosynthetic materials, proline content, relative water content (RWC), etc. Improved plant tolerance and thereby supported the development of plants.	[112]
*Trichoderma* spp.	*Solanum lycopersicum* and *Zea mays*	These fungi are parasitic and saprophytic, residing in the rhizosphere. They help plants overcome abiotic stresses, such as cold, drought, heat, and salinity	[113]
*Piriformospora indica*	*Trigonella* *foenum-graecum*	The fungus revealed positive effects in the mitigation of salinity stress in fenugreek plants and improved various growth responses	[114]
AMF	*Triticum aestivum*	AMF controls the plant proteome under field conditions. The interaction of bacteria and fungi revealed proteins employing STRING that interact with various proteins to partake in seed development and toleration of abiotic factors	[115]
*Funneliformis geosporum*, *Rhizophagus irregularis*, and *Claroideoglomus claroideum*	*Cicer arietinum*	The introduction of PGPF and an artificial supply of water at a vital level improved chickpea growth and produced an abundant grain harvest compared to uninoculated plants without water stress	[116]
AMF	Medicinal and aromatic plants	The fungi can help mitigate abiotic environmental stresses, such as water stress, salt stress, and low and high temperatures	[117]
*Saccharomyces cerevisiae*	*Arabidopsis thaliana*	The fungi are known as biostimulants and can assist crops in withstanding abiotic stresses, such as drought, salinity, or cold	[118]
*Serendipita indica*	*Zea mays*	The fungi improve plant growth management under abiotic stress conditions	[119]

#### PGPF Potential against Heavy Metal Contamination

Some abiotic factors, including heavy metal contamination, climate change, drought stress, extreme temperature, etc., affect the development of PGPF and the mycorrhizal association with crop plants. In most cases, the presence of these microorganisms can reduce stresses prone to crop plants due to these abiotic factors. We aimed at the potential of heavy chemical concentrations in the soil to cause pollution., such as ineffective mining activities, handling of sewage sludge, other industrial effluents released into the surrounding soil, and misuse of heavy-metal-containing fertilizers on farmlands. Other activities by which heavy chemicals accumulate in the environment include gas exhausts, among other factors, which lead to the accumulation of heavy metals, organic pollutants, and radionuclides in the soil and surrounding environment. This toxicity causes harm to human health and the ecosystem at large [120] and cannot only be treated strictly or levity, but instead should be attended to in a way that can modify the potential of any danger due to the pollution of the food chain and nearby rivers by wind erosion. The induction of microbial communities inhabiting the rhizosphere by plant roots has been brought to notice. The potential of microbial communities is a significant factor, contributing to the solubility of metals and as immobilizers of soil metals as a result of hydrated ferric oxides and the precipitation of sulfides or complex sugar (polysaccharides) exudation [121]. On the surface of fungal cell walls, the functional groups perform a significant task in the adsorption of metals from the soil [122]. Considering other features, Cu, Pb, and Zn can be obtained from oxides or carbonates through the potential microbial communities [123]. Pollutants can be eradicated from the soil environment via biological means by using PGPF, PGPR, and Arbuscular mycorrhizal (AM) [48]. The symbiotic associations of AM or PGPF with crop plants improve the evolution of plants in heavy-metal-contaminated areas by the following processes: phytohormone production, pathogenic resistance, nutrient acquisition, water relations, amelioration of the soil structure, and contribution to soil aggregation and therefore promote bioremediation processes. For example, PGPF and AMF have been reported to reduce caesium assimilation by plants [124] and therefore could be employed in the production of plantations on soil contaminated with radionuclides and so possess the ability to decrease danger caused to the environment. PGPF and AMF also possess the ability to monitor the toxicity of the soil [16] and its effectiveness. The application of AMF features as a sign of changes happening during soil restoration and a tool for biological observation of the soil quality. The stages of colonization of grasses in polluted soil fields have been revealed to correspond with heavy metal contamination [125].

### 2.3. Arbuscular Mycorrhizal Potential in the Rhizosphere of Crop Plants

Arbuscular mycorrhizal fungi (AMF) have been reported to contribute to nutrient uptake in plants and tolerate abiotic stress [24]. AMF commence a symbiotic association with the host plants they inhabit and regulate the development of crop plants. The network of AMF mycelial broadens under the roots of crop plants, promoting nutrient uptake [126]. The common mycorrhizal network (CMN) has a greater effect on plants on the fungal-mediated transport of N and P and thereby promotes the development of crop plants following strenuous states of the environment [24]. AMF have also displayed their adaptation in plants when exposed to various conditions of habitats, especially those affected by abiotic stresses, such as drought, chilling, salinity, high temperature, heavy metals, and flooding [127]. It has been reported that two forms of fungi are implored in stress tolerance, and they are AMF and ectomycorrhiza fungi (EMF), reported by Koza, et al. [24]. AMF dwelled in the host plants without effecting danger and were known to elicit the toleration of various stresses. Redman, et al. [128] explained how the toleration of abiotic stresses could be stimulated via utilizing the endophytic AMF that survive in mutual associations with about 80% of plants. Furthermore, AMF have been reported to contribute to the improvement of salt tolerance, employing various methods which constitute regulating the rate of K^+^/Na^+^ in plant cells, the production of growth hormones, the development of the rhizosphere and other soil conditions, the transfer of ion salts to the vacuoles, and, likewise, the promotion of photosynthetic water use effectiveness [129]. Furthermore, AMF improve salt stress activities by improving the amount of sugar and electrolytes and thus perform the role of osmoregulation [130]. Another investigation showed how AMF stimulate the tolerance to salt stress by upregulating the antioxidant potential via activation of the plant glutathione–ascorbate cycle. The symbiotic association of AMF has also been revealed to promote salt resistance in crop plants, such as tomatoes, lettuce, clovers, cucumbers, and maize [130]. AMF-stimulated tolerance has been revealed as a usual characteristic of abiotic stress tolerance in drought-stressed plants.

## 3. Crop Plants Establish Beneficial Microbes in the Rhizosphere

The roots of the crop plants secrete primary metabolites, such as amino acids, carbohydrates, organic acids, glucosinolates, and vitamins, and secondary metabolites, including alkaloids, flavonoids, phenolics, sulfur-containing compounds, and terpenoids, which are known to build, signal, and interact with the rhizosphere microflora. This exudation in the soil at the root region is made up of an abundant mixture of chemicals obtained at an important rate of carbon and nitrogen for the crop plants’ growth and health status. The exudates have a greater advantage of attracting and increasing the beneficial microbial communities, while destroying pathogenic organisms by inhibiting their growth and invasion [131].

The rhizosphere microbiota likewise help plants to adapt to their specific environment, and the inhabitation of specific microbiomes in the rhizosphere is known as niche colonization [132]. The significance of the rhizosphere microbiome, otherwise known as rhizobiome, is considered to depend on the chemical exudates, which also promote mutual activity through signaling molecules that are stimulated and secreted by plants and microbes. Scientific investigations have focused on PGPR but are turning to the potential of PGPF and protists that give room for the production of more rhizobiomes in the future. Plant signaling and the impact of the rhizobiome chemistry are still understudied. Researchers sometimes refer to this point by employing a lot of terminologies such as complex plant–microbe interactions, rhizosphere chemical language, rhizosphere interactions, and plant signaling [133].

The recruiting and colonization of the rhizobiome by the rhizosphere chemistry can occur following two general processes. First, through induction by the rhizodeposits or root exudates produced by microbial biomass inhabiting the rhizosphere of the crop plants [134]. This is a special route for shaping, recruiting, and tuning microbial communities from the source of microbes in the soil, constituting the methods that disallow, support, or end microbial multiplication. However, we believe that this crucial function of chemicals in investigating rhizosphere residents cannot be regarded as signaling.

The second process affecting the rhizobiome takes place through the sensing and response to plants or microbe compounds of low molecular weight [135], leading to a cellular reaction(s) that is curtailed by the substance being perceived through the following processes: catabolism, transformation, resistance, etc. This constitutes a restrictive cascade that eventually results in the transcription of loci in reaction to a specific compound. The major type of signaling is divided into three mechanisms known to take place in the rhizosphere, and they are:

Quorum sensing (QS)—microbial intraspecies and interspecies signaling that allows for cell-to-cell signaling or communication, i.e., PGPR or PGPF, to share information about cell biomass or density and regulate gene expression. The signal molecules permit microorganisms to multiply and modify their behavior;

Signaling from plants to microorganisms through plant-secreted materials that have been revealed in quite a lot of specialized symbiotic associations and most probably take place often in other interactions;

Signaling from microbial communities to plants referenced so far by compounds produced by microbes participating in plant defense responses, plant gene expression, and root development.

### Plant Signaling and Impacts on Rhizobiomes Contributing to Crop Plants’ Growth

The association of crop plants and PGPF involves the interchange of signal molecules with each other. This first interchange forms the identification of the befitting partner and carries out an integral function in producing a successful association. Plant reactions to the microbial relation change into greater changes in physiological state, biochemical reactions, and metabolic adjustments [136]. Due to modern advancements in molecular biology in the beneficial association of plant–microbe interactions, some factors of the signal transduction pathways have been characterized. Therefore, plant signaling pathways result in improved plant growth due to PGPF depending on endogenous regulators, such as auxin, ethylene (ET), and cytokinin (CK). Gibberellin (GA) and abscisic acid (ABA) are other phytohormones representing other forms of signaling components that promote the interactions of beneficial plants and PGPF (Table 3).

The association of crop plants and PGPF can use direct or indirect methods: either the plant organs (roots) in the rhizosphere soil or above the soil (shoots). The reported effects are the development of root hairs, improved biomass production, flowering, and increased production. A study reported how *T. virens* introduced into wild-type Arabidopsis seedlings revealed biomass production and the development of lateral roots [137]. The inoculated plants demonstrated the activity of auxin-regulated genes. Despite mutations anticipated to occur in the genes that take part in auxin signaling, *AUX1*, *AXR1*, *BIG*, and *EIR1* inhibited the plant-growth-promoting and root developmental characteristics of *T. virens* introduced to Arabidopsis. The investigation showed that the plant growth promotion by *T. virens* functions via the standard auxin effect pathway. Similarly, another study revealed the effects of treatment with IAA; thus, IAA was noted to relieve the spoiled root hair phenotype of the *rhd6* mutant [137]. The results obtained showed that microbial auxin may contribute to inhibiting the root hair formation defects of *rhd6.* So, auxin is a major signaling component in the association of plant and microbial communities.

Another phytohormone that is essential for plant growth and development is ethylene (ET), which occurs in a gaseous state. This phytohormone also responds to environmental signals [138].

Barazani, et al. [139] reported how the *N. attenuata* plant was changed to silence ET stimulation; improved plant growth activity by the fungus was not obtained. The differential stimulation of genes associated with ET biosynthesis and signaling in the roots of barley plants inhabited by the endophytic fungus *Pi. indica* was revealed by a DNA microarray-based gene expression procedure [140]. Mutant strains *etr1*, *ein2*, and *ein3/eil1* were destroyed in the ET signaling pathway and demonstrated a reduction in growth and seedling responses by this fungus, unlike the wild type. This showed the potential of ET signaling in the advantageous association between the two symbionts [141]. So, the proposition that ET signaling contributes to plant growth by fungal inhabitation on barley plant roots is obvious.

Some volatile organic compounds (VOCs) produced by PGPF otherwise control the development of crop plants, and the signaling pathways interceding VOC sensing are not fully apprehensible. From *Trichoderma* spp., the prima biological antifungal VOC isolated was 6-pentyl-2H-pyran-2-one (6-PP) [142], which stimulates *A. thaliana* root growth through auxin signaling and the ET response modulator EIN2 [143]. Hashem, et al. [144] proclaimed that cytokinin (CK) signaling performs specific functions in growth improvement when subjected to *Bacillus subtilis* GB03 VOCs. CKs are significant for the stimulation of *Pi. indica* growth improvements in Arabidopsis [145]. Furthermore, in effect to *Pi. indica* establishment, the ABA pathway was suggested to improve plant growth through protein kinases, cellular [Ca^2+^] increments, and phosphoinositides [146]. The manufactured additional phytohormones by PGPF are gibberellin (GA) and brassinosteroids [147]. On a short note, phytohormone signaling shows how it participates in producing compatible associations (symbiosis) between PGPF and plant hosts, which lead to growth improvements and abundant biomass.

**Table 3 jof-09-00239-t003:** PGPF mechanisms in relation to crop plant growth and development.

Mechanisms	PGPF	Activities	References
Phytohormone production	*Actinomucor elegans* and *Podospora bulbillosa*	Promote the ability to withstand water deficiency and salinity stresses in tomato plants	[56]
	*Trichoderma aureoviride* TaN16, *Penicillium citrinum* PcK10, and *Aspergillus niger* AnK1	These PGPF possess the potential to produce IAA that contributes to the growth of the plant	[148]
	*Penicillium* spp., *Clonostachys* spp., *Trichoderma* spp., *Purpureocillium* spp., *Aspergillus* spp., *Taifanglania* spp., and *Trichoderma* spp.	The mentioned strains can produce siderophores and IAA thereby contributing to plant growth from leguminous–manure-incorporated soil and non-leguminous–manure-incorporated soil	[52]
	*Cladosporium cladosporioides*, *Penicillium simplicissimum*, and *Cladosporium pseudocladosporioides*	The PGPF strains were reported to produce IAA and siderophore hormones	[49]
	*Aspergillus fumigatus*	The fungus was reported to produce IAA to enhance the growth of rice plants	[149]
Phosphate solubilization	*Trichoderma harzianum TaK12* and *Trichoderma aureoviride* TaN16	The PGPF contribute to the growth of rice plants by solubilizing phosphorus	[148]
	*Penicillium* spp., *Trichoderma* spp., *Purpureocillium* spp., *Taifanglania* spp., and *Aspergillus* spp.	The mentioned PGPF involve phosphate solubilization in green and non-green manure soil	[52]
	*Penicillium chrysogenum*	The strain was recorded to solubilize phosphate in wheat growth under low nitrogen input	[150]
	*Cladosporium cladosporioides* and *Penicillium simplicissimum*	The fungi solubilize phosphate in *Vicia villosa* Roth	[49]
	*Penicillium* spp.	Solubilize phosphate	[151]
Inhibition of phytopathogens	*A. flavus*, *Mucor circinelloides*, *A. niger*, and *P. oxalicum*	The PGPF displayed great potential in the inhibition of phytopathogens in tomato plants	[15]
	*Chaetosphaeronema achilleae*, *Acrophialophora levis*, and *Penicillium chrysogenum*	They were reported to suppress the growth of the phytopathogen *Alternaria alternate* in wheat plants	[150]
	*Trichoderma viride* and *Penicillium chrysogenum*	Biocontrol spoilage organism of orange fruits	[10]
	*Pyrenophora* spp., *Sordaria* spp., and *Penicillium* spp.	The fungi were reported to be present in the rhizosphere of tomato plants, improving the health status of tomatoes by inhibiting phytopathogens	[7]
	*Aspergillus flavus*	The fungus produces biocontrol agents, acetate, linalool, linalyl geranyl acetate, oleic acid, 1-eicosanol, and 1-chloro-octadecane, that inhibit the growth of phytopathogenic fungi on *T. foenum-graecum*, *S. lycopersicum*, *P. oleracea*, and *L. sativum*	[152]
Mitigation of abiotic stress	*Cunninghamella bertholletiae*	The fungus was reported to inhibit abiotic stresses, heavy metal, drought, and salinity, in tomato plants	[109]
	Arbuscular mycorrhizal fungi (AMF)	Carry out an important role in alleviating abiotic stresses	[110]
	Arbuscular mycorrhizal fungi (AMF)	Mitigate the abiotic conditions of crop plants	[24]
	__	The PGPF mitigated drought stress affecting crop plants	[153]
	*Penicillium olsonii*	The fungus was acquired from the rhizosphere of *A. littoralis*, which increases plant growth and tolerates salinity and extreme temperatures	[154]
Volatile organic compounds (VOCs)	*Trichoderma harzianum*, *T. Hamatum*, and *T. velutinum*	These PGPF have been reported to produce VOCs to induce plant responses	[155]
	*Trichoderma* spp.	VOCs in biological control agents *A. panax*, *B. cinerea*, *C. destructans*, and *S. nivalis* and promotes plant growth	[156]
	*Trichoderma* spp.	VOCs produced by PGPF, together with PGP activity, can control spoilage organisms in the rhizosphere by antifungal and antibacterial potentials and have long-range control potentials as a result of their volatile nature	[157]
	*Trichoderma asperellum*	The fungus produces antifungal and antibacterial agents such as xylanase, cellulase, pectinase, protease, and chitinase	[158]

## 4. Formulation of PGPF, Their Application, and Their Effect on Crop Plants

### 4.1. Formulation of PGPF

PGPF, most importantly *Trichoderma* spp., have many advantages as a plant-growth-promoting agents and are known to have a lot of uses in commercial formulations. Different organic and inorganic carrier molecules have been investigated for efficient features of bioinoculants. From *T. harzianum*, a talc-based formulation was produced to provide concentrated biomass of conidial of the fungus with increased colony forming units (CFU) for a long time [159]. The concentrated formulation made available the added benefit of smaller packaging for keeping and conveying, and low product cost as compared to other carriers, such as charcoal, cow dung, sawdust, and vermiculite. The application of seeds for the formulation revealed absolute improvements in promoting growth in chickpeas [160].

Corn, cassava peel, ground nut husks, and sugarcane bagasse have been employed as carriers for *Trichoderma* sp. inoculants [161]. The corn formulation produced a drastic improvement in the root length of rice seedlings, wet weight, and biomass, unlike the inoculum introduced with sugarcane bagasse and the control [162]. A formulation in the form of flowable powder was produced for biostimulant *Trichoderma* strain dry sprays employing a CO_2_-producing dispersant system based on citric acid, sodium bicarbonate, and polyacrylic acid with lecithin and polyvinyl alcohol employed as adhesives and the former as a wetting agent [163]. Hydrolytic amino acids have also been reported to be derived from dead pigs to prepare *T. harzianum* and used as biofertilizers. The biofertilizer produced contributed to higher densities of *T. harzianum* and promoted plant growth when utilized as a soil improvement. Compost manure was obtained from cow dung from which *Trichoderma* biofertilizers were manufactured and applied to the agricultural field [164]. The field in which the biofertilizer had been applied displayed bountiful harvests and was more productive than agricultural soil in which biofertilizers were not introduced [165]. Another efficient formulation was produced from *Piriformospora indica*, which was manufactured from talcum powder (vermiculite) with 20% moisture. In glasshouse experiments, these formulations carried out substantial improvements as bioinoculants compared to vermiculite-based formulations [166]. These reveal the possibility for biofertilizers formulated from PGPF to contribute to the development of the agricultural system as well as improve crop productivity.

### 4.2. Application and Effect of PGPF on Crop Plants

Microbial communities coexist in inter-relationships with the roots of crop plants. Plant-growth-promoting microorganisms (PGPMs) have been employed in the stimulation of resistance in plants against pathogen invasion aside from promoting the development of crop plants [7]. The introduction of these microbes is one of the ecologically friendly management methods of disease that is durable as a result of the stimulation of innate immunity in plants. Few studies have explained how PGPF, in comparison to PGPR, obtain their effectiveness in the stimulation of resistance against the invasion of pathogens [167,168,169]. PGPF are nonpathogenic, they occur naturally as saprophytes that feed on dead organic matter, they assist in maintaining soil fertility, and as a result, they promote plant growth and stimulate the defense response against phytopathogen attacks [16]. The potential of PGPF to colonize roots is regarded as the first mechanism required in preventing phytopathogen invasion and likewise assists in the assimilation of nutrients, therefore improving plant growth [16]. The potential of PGPF involves phosphate solubilization, IAA production, siderophore production, enzyme (cellulase and chitinase) production, etc., either directly or indirectly, to promote plant growth, apart from stimulating disease resistance [24].

As a matter of the advantageous value of PGPF in the agricultural system, scientists have drawn their concentration on the employment of PGPF for stimulating inhibition via ISR and promoting plant growth in crop plants [170]. ISR produced by PGPF in plants takes part in modifying cell walls via callose, lignin, and phenol accumulation, which reduces their income and also prevents the growth and multiplication of the intruding phytopathogens [151]. Aside from changes that occur in the cell walls of plants, PGPF also initiate the improved collection of defense-associated enzymes in crop plants (β-1,3 glucanase, chitinase, peroxidases (POX), phenylalanine ammonia-lyase (PAL), etc.) that are proportionally associated to protective methods against phytopathogens [171].

## 5. PGPF as an Alternative to Chemical Derivatives in Crop Plantations

Chemical derivatives, including fungicides, herbicides, and pesticides, can be classified based on their applications to combat organisms causing disease, invasion of plants, and their chemical nature. These derivatives are used in agriculture to confer protection to crops from crop pests, insects, and unwanted plants (weeds). The chemicals are also applied in public health to eradicate disease, causal vectors, and to kill insects such as cockroaches, mosquitoes, aphids, termites, etc. The chemical derivatives are classified into different groups based on their potential action. For example, chemical derivatives targeting insects are insecticides, those targeting fungi and various mycotoxins produced by fungi are fungicides, those applied to remove weeds are herbicides, and those used against rodents (rats) are called rodenticides. Those employed to control the existence of insects such as bugs, aphids, and other pests are fumigants, and insect repellents are mostly applied on the skin and/or clothing to prevent insects. Based on the nature of chemicals, they can be classified as benzoic acid derivatives, benzonitriles, carbamates, dipyrids, organochlorines, organophosphates, phenoxyalkonates, phenyl amides, phtalimide derivatives, trazines, and pyrethroids [172].

As much as these chemicals are effective in carrying out their primary assignment of eradicating unwanted organisms, they also pose a threat to crop plants, man, animals, and the environment they are applied to, thereby causing various hazards and depriving healthy conditions because of the toxic accumulation in the tissue of crops, man, and animals feeding on them [173]. For this reason, scientists have diverged to creating affordable and ecofriendly means of controlling phytopathogens and various plant diseases.

PGPF have demonstrated various antagonistic activities against all forms of phytopathogens and diseases of plants. Aside from their antagonistic potential, they also support the growth of such plants and act against abiotic conditions (Figure 2, Table 2) that can hamper the healthy status of plants. Antagonism, referred to as ‘biocontrol’, is a process by which living organisms are employed to mitigate or inhibit the growth of a phytopathogen population [174]. These microorganisms are ecofriendly in nature, i.e., they do not cause harm to man or animals while consuming the crops and the environment they live in. The studies discussed within the context showed that *Trichoderma* spp. has been used as a biocontrol agent in controlling various phytopathogens of crop plants. These organisms are primarily found in the rhizosphere region of healthy plants, where they control the invasion of disease-causing organisms (Table 1). Rojas-Sánchez, et al. [124] explained how PGPF dwells in root cells, root surfaces, and the outer layers of root cells. De Palma, et al. [157] reported how *Trichoderma* spp. can be introduced into plant roots during the seedling stage of plant growth to ensure a beneficial relationship in improving plants’ health and preventing disease invasion. The basic procedure for the biocontrol potential of *Trichoderma* spp. against pathogens is as follows: identification and colonization of plant pathogenic fungal species via disruption of the pathogenic cell wall and assimilation of secreted nutrients, referred to as mycoparasitism [16], eliciting the resistance against diseases by altering the plant roots during confrontation with the phytopathogens [175], and inhibition of root-knot disease and cyst nematodes by eliminating the nematode eggs, juvenile stages, and adult nematodes.

The formulation products of *Trichoderma* spp. are produced in granules or a wettable powder. Majorly, 90% of the strains of *Trichoderma* spp. are employed in crop protection to control plant diseases through antagonistic features elicited by the fungi against phytopathogens [176]. The potential of their application as a BCA on farmland is estimated on the input cost and proportion to the production of crops. The application of BCA is not expensive and is easy to handle, unlike synthetic chemical derivatives [177]. Farmers can make use of synthetic chemicals because of their low cost, but they cannot avail abundant harvests and healthy crop production similar to BCA. So, they observed huge economic losses when crop production and input costs were not balanced. Therefore, *Trichoderma* spp. and other PGPF are known for enhancing crop productivity, which amounts to an improvement in revenue. The appropriate application of BCA (PGPF) on farmland can reduce and substitute chemical fertilizers. To maintain soil health, the application of PGPF has contributed to a sustainable approach. Scientists have studied how PGPF inhibit the growth of phytopathogens, as discussed in this review.

## 6. Conclusions

PGPF and their associations are deliberately different, and various factors contribute to their plant health activities. The scientific results express the importance of rhizospheric plant-growth-promoting fungi that reveal the potential to favor crop plant growth and development. The application of PGPF bioformulations has helped reduce the problems encountered in the production of crops and the endeavor of pest-free crops with no or minimal applications of chemical fertilizers and pesticides. This review portrays how the rhizospheric microbiome (PGPF) can improve the sustainability of agricultural products and soil fertility. However, comprehensive studies based on the attributes of PGPF contributing to the growth of crop plants and productivity are still in their infancy. So, it is important to examine some of the activities carried out by PGPF in plant–microbe interactions for incorporating microbiota in the improvement of sustainable agriculture for crop production and protection.

Recent progress in biotechnological implements and dependable change could be useful in the production of the PGPF to bestow beneficial attributes to crop plants. Genetic change and upregulation of plant-growth-promoting traits that perform synergistic activities may produce improved functions by the inoculant. Studies can be employed periodically to measure the genetic stability and ecological preservation of the genetically modified strain. Therefore, PGPF reveal potential in contributing to plant growth via the extensive improvement of basic process operating industries.

## Figures and Tables

**Figure 1 jof-09-00239-f001:**
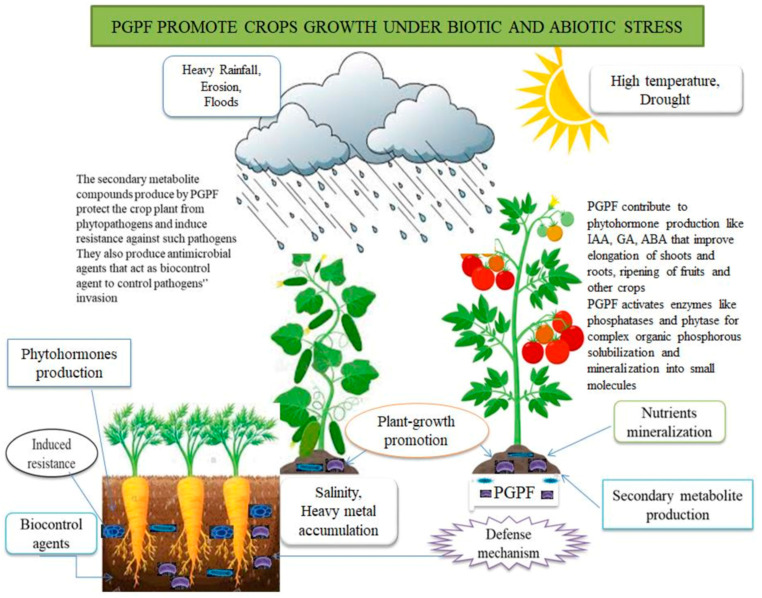
The plant-growth-promoting fungi biotic and abiotic factors potential of crop plants.

**Figure 2 jof-09-00239-f002:**
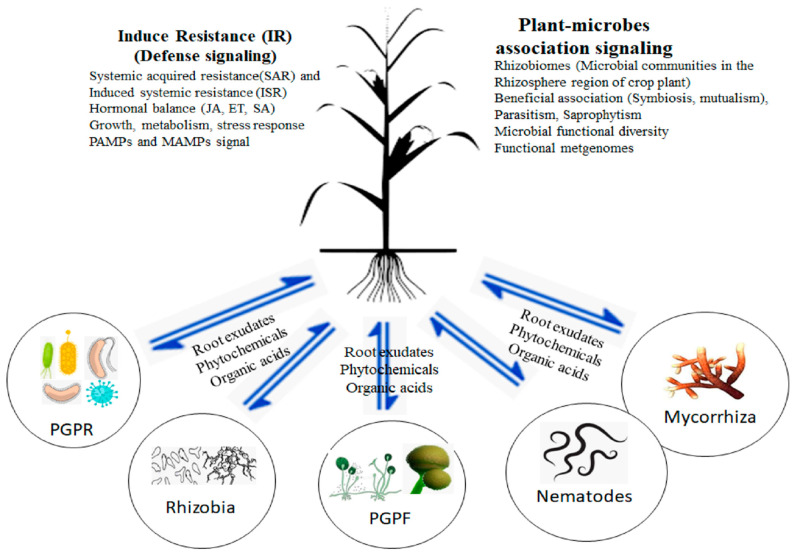
PGPF and other microbial communities induce resistance and initiate defense signaling in crop plants.

**Figure 3 jof-09-00239-f003:**
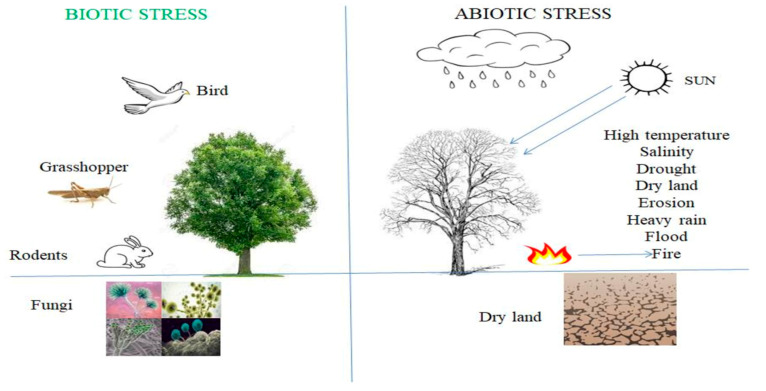
Plant-growth-promoting fungi affect crop production under biotic and abiotic stresses.

**Table 1 jof-09-00239-t001:** Plant-growth-promoting fungi contribute to crop plant growth and development.

PGPF	Crop Plant	Activities	References
*Aspergillus flavus*, *Aspergillus niger*, *Mucor circinelloides*, and *Pencillium oxalicum*	*Solanum lycopersicum*	Biological control activity of PGPF against *F. oxysporum*	[15]
Arbuscular mycorrhizal, *Trichoderma* species	*Artemisia annua*, *Arabidopsis*, *Zea mays*, *Oryza sativa*, *Arachis hypogaea*, *Helianthus annuus*, and *Solanum lycopersicum*	Disease combating, detoxification of organic and inorganic toxic chemicals, induction of systemic resistance, plant growth promotion, secretion of secondary metabolites, and heavy metal tolerance ability	[48]
*Daldinia eschscholtzii*, *Sarocladium oryzae*, *Rhizoctonia oryzae*, *Penicillium allahabadense*, and *Aspergillus foetidus*	*Cucumis sativus*	Promote plant growth by stimulating secondary metabolites, including phytohormones, siderophore, and phosphate-solubilizing metabolites	[19]
*Trichoderma*, *Rhizoctonia*, *Fusarium*, *Penicillium*, *Talaromyces*, *Gliocladium*, *Phoma*, and *Phytophthora*	*Zea mays* and *Oryza sativa*	Crop protection and crop yield promoting seed germination, enhanced root and shoot growth, and producing fruit and chlorophyll.	[41]
*Cladosporium cladosporioides*, *Penicillium simplicissimum*, and *Cladosporium pseudocladosporioides*	*Vicia villosa*	Ability to promote plant growth and biocontrol effects against *Calonectria ilicicola* in plants	[49]
*Alternaria tenuissima*, *Byssochlamys spectabilis*, *Nigrospora chinensis*, *Cephalotheca oveolate*, *Chaetomium globosum*, and *Penicillium melinii*	*Aeluropus littoralis*	Biostimulant activities	[50]
*Aspergillus niger*	Forage grass	Production of IAA, siderophores, ammonia, phosphate solubilization, 1-aminocyclopropane-1-carboxylate (ACC) deaminase, and enzymes such as proteases, phosphatases, and other hydrolases	[51]
Leguminous (*Clonostachys* spp., *Trichoderma* spp., and *Penicillium* spp.) and non-leguminous (*Purpureocillium* spp., *Taifanglania* spp., *Trichoderma* spp., and *Aspergillus* spp.)	*Vicia villosa* (leguminous) and *Brassica juncea* (non-leguminous)	Promote or enhance plant growth. The strain solubilizes phosphorus and produces a siderophore, while others revealed the potential to produce IAA with/out tryptophan. Extracellular enzyme potentials, including endoglucanase and β-glucosidase activities, were also confirmed in the soil-incorporated green manures	[52]
*Trichoderma* spp, and *Aspergillus* spp.	*Cucumis sativus* and *Arabidopsis*	Contribute to biofertilizer stimulation with soil resident fungi, thereby improving plant growth	[53]
*Trichoderma longibrachiatum*	*Triticum aestivum*	Promote plant growth and induce immunity to parasitic nematodes	[54]
*Trichoderma* sp., *Penicillium* spp., *Aspergillus* spp., *Phoma* spp., *Fusarium* spp., *Aspergillus* spp., *Chaetomium* sp., *Metarhium* spp., and *Acremonium* spp.	*Zea mays*	improve plant health and contribute to the growth and development	[55]
*Actinomucor elegans* and *Podospora bulbillosa*	*Solanum lycopersicum*	The fungi produce higher amounts of chlorophyll, antioxidants, amino acids, carotenoids, proteins, activities, sucrose contents, glucose, salicylic acid, and fructose and reveal hydrogen peroxide and lipid metabolism relative to influence plant growth.	[56]

## Data Availability

Not available.

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
