# Peer review of "Fungi That Promote Plant Growth in the Rhizosphere Boost Crop Growth"

_jof, 2023, doi:10.3390/jof9020239_

Round 1

Reviewer 1 Report (Previous Reviewer 2)

Figures are blurred make sure it is 300 dpi. Do this for all the figures.

The manuscript is much improved.. Good work..

Author Response

Point 1: Figures are blurred make sure it is 300 dpi. Do this for all the figures

Response 1: All the figures have been converted in TIFF format 300 dpi and uploaded as a zip file in the provided tab.

Reviewer 2 Report (New Reviewer)

Dear Author,

Overall manuscript is well written.

1.       What is the main question addressed by the research?

Comments: The main focused of current manuscript is describing the role of  PGP Fungi as an effective bioagent to facilitate and promote crop production, plant growth, resistance to disease invasion, and various abiotic stresses.

2.       Do you consider the topic original or relevant in the field? Does it
address a specific gap in the field?

Comments: The topic is not original there were several paper published in same and other journal but the current manuscript compiling the latest information regarding the PGPF.

3. What does it add to the subject area compared with other published
material?

Comments: Most of the paper published focused on bacteria base PGP but the current one compiling the fungal based PGP.

4. What specific improvements should the authors consider regarding the
methodology? What further controls should be considered?

Comments: No methodology used for the paper. Author compiled the information.

5. Are the conclusions consistent with the evidence and arguments presented
and do they address the main question posed?

Comments: Author clearly concluded the topic in his paper.

6. Are the references appropriate?

Comments: Yes.
7. Please include any additional comments on the tables and figures.

Comments: -Figure 3 in very basic either improve or merge with figure 1. for biotic and abiotic stress make one common figure.

there were few comments

-Add few examples of fungi in abstract.

-Figure 3 in very basic either improve or merge with figure 1. 

- for biotic and abiotic stress make one common figure.

-Conclusion must be focused on future prospects of the work. 

Author Response

Point 1: Add few examples of fungi in abstract.

Response 1: Examples of PGPF have been mentioned in the abstract of the manuscript

Point  2: Figure 3 is very basic either improve or merge with figure 1.

Response 2: the figures have been merged in the new figure 1

Point 3: for biotic and abiotic stress make one common figure.

Response 3:  new figure has been inserted for biotic and abiotic stress on plants

Point 4: Conclusion must be focused on the future prospects of the work.

Response 4: The future prospect has been added

This manuscript is a resubmission of an earlier submission. The following is a list of the peer review reports and author responses from that submission.

Round 1

Reviewer 1 Report

Thanks for your work that reviews the use of PGPF and plant growth. I don't know if your work is innovative. In the title, you talked about PGPF but needed to cite mycorrhiza and the effects.

You should improve the part about the formulation, applications, and   effects on plant growth (there is a lack of information about positive effects on plant production, like fruits and biomass). I don't understand the aims

Reviewer 2 Report

The manuscript entitled “Rhizosphere plant-growth-promoting fungi enhance the growth of the crop plants by Adedayo and Babalola.

The review is informative and novel, focused on PGPF and its role in promoting the growth and development of crop plants. Here are my comments and suggestion for this submission.

 Comments

·       The introduction is not enough; the title says that PGPF enhances the growth of crop plants, but no other crop plants are discussed in the introduction except tomato. So, the literature cited in the introduction is not well enough.

·       Lines 78-79 it is written as limited studies are reported about PGPF; if so, they covered most of the crop plants because the title is broad.

·       In lines 39-40, the sentence is unclear; please elaborate on or rewrite it.

·       It would be better if you could have categorized the review into biotic and abiotic parts, which can help the reader.

·       Regarding biotic stress, the authors are well-presented about induced resistance, defense mechanism, defense signalling, and biochemical defence. But my suggestion is to make a pictorial explanation by explaining how PGPF induce the resistance and how they initiate the defense signalling.

·       The figures are very simple, especially fig2, remove the shadings of letters in figure 1 because the text is unclear. In figure 2, salinity is mentioned twice, so remove one (in the orange box).

·       As written in lines 222 and 496 (conclusion part), I am interested to know that there would be a reduction in chemical pesticide usage if we could use these PGPF as biological control organisms.

·       Line 333 and 362 alignments are missing for the subheading.

·       I was much more curious to see the action of PGPF against heavy metal contamination (line 385), but there is not much discussion on this.

·       Most of the abiotic section is just reporting and citing the reference.

The manuscript is recommended for publication after addressing the comments.

Reviewer 3 Report

The present review paper "Rhizosphere plant-growth-promoting fungi enhance the 2 growth of the crop plants" compiled by authors is a very basic and cannot be accepted in this form in such a reputed journal. The review paper contains very basic point, which are  very common  and  found in every  even book chapters.

Now a days significant progress have been recorded in the rhizosphere research, molecular plant -microbe interactions, signaling molecules , how the plant recruits beneficial microbes but author had not covered any these aspect.